# Towards a Risk-Based Analysis of Terms of Service for Consumer Decision Support

Elena Molino-Peña[1,2], Beatriz Esteves[3], José María García[1,2] and Antonio Ruiz-Cortés[1,2]

[1]*Smart Computer Systems Research and Engineering Laboratory (SCORE), Universidad de Sevilla, Seville, Spain*
[2]*Research Institute of Informatics Engineering (I3US), Universidad de Sevilla, Seville, Spain*
[3]*IDLab, Department Electronics and Information Systems, Ghent University – imec, Ghent, Belgium*

### Abstract
The definition and monitoring of the parties' responsibilities established in the Terms of Service (ToS) of Software as a Service (SaaS) distribution models are usually under the control of the provider, which introduces risks of power asymmetry and lack of transparency. Many of these clauses are not operationalizable or verifiable in an unequivocal manner, making it difficult for customers to determine whether an obligation has been fulfilled, a right can be effectively exercised, or a prohibition has been violated. This complexity increases in multi-service and decentralised architectures, such as Solid pods, where responsibilities and verifiable evidence are distributed across multiple actors and systems. In this paper, we explore the risks to which customers are exposed when subscribing to cloud services and propose an initial approach to automatically infer these contractual risks through the analysis of potentially unfair terms contained in the ToS.

### Keywords
Verifiable Contractual Clauses, ODRL, Terms of Service, Contractual Risk Detection

## 1. Introduction

Terms of service (ToS) are adhesion contracts unilaterally drafted by the provider that are used to define the provisions and conditions under which services are provided, including the rights and responsibilities of the parties, renewal, changes, indemnities, and termination terms, among other legal aspects [1]. However, are all these terms verifiable? Checking whether obligations, rights, and prohibitions are complied with remains a significant challenge, particularly when customers cannot independently determine whether the provider is complying with its own commitments. In such situations, the provider effectively retains discretionary authority to determine whether compliance has occurred, which may in itself constitute an unilateral unfair conduct.

The monitoring of contractual terms, especially those that are non-verifiable, tends to remain under the control of the provider or, in the event of a dispute, the body responsible for resolving the case [2]. A similar form of contractual opacity may arise in Solid architectures, where ToS continue to govern the provision of infrastructure, applications, and related services [3]. The lack of mechanisms increases the risk of consumer vulnerability, reduces transparency, and increases power imbalance. When such clauses are also potentially unfair, meaning that they create a significant imbalance in the parties' rights and obligations to the detriment of the consumer [4], they become a source of contractual risk [5].

By risk we understand *"a possible event that could cause harm or loss, or make it more difficult to achieve objectives."* [6]. From the perspective of the Fitness Check on Digital Fairness [5], ToS frequently operate as sources of contractual risk for consumers, particularly when they contain unfair terms. These risks may materialize in the form of financial harm, data loss, legal/jurisdiction associated risks, lack of awareness of the contractual conditions, uncertainty caused by unclear terms, privacy harm, and difficulties in exercising consumer rights. A clear structural imbalance emerges in this context: the less risk assumed by the service provider, the more risk is effectively transferred to the consumer,

*4th Solid Symposium, London, 30 April & 1 May 2026*

✉ mmolino@us.es (E. Molino-Peña); beatriz.esteves@ugent.be (B. Esteves); josemgarcia@us.es (J. M. García); aruiz@us.es (A. Ruiz-Cortés)

🆔 0000-0001-7024-5300 (E. Molino-Peña); 0000-0003-0259-7560 (B. Esteves); 0000-0002-0303-2740 (J. M. García); 0000-0001-9827-1834 (A. Ruiz-Cortés)

an asymmetry that produces a substantial economic impact, with the annual detriment suffered by consumers estimated at approximately €7.9 billion [5].

To address this problem, this paper explores the feasibility of using Open Digital Rights Language (ODRL)/Terms of Services Language (TOSL) to formally represent contractual terms and to infer associated risks using the Data Privacy Vocabulary (DPV), with the aim of enabling the automatic identification of risks posed to consumers by potentially unfair contractual clauses. We propose the use of a rule-based reasoning engine (an N3 reasoner) to classify abusive contractual terms expressed in ODRL/TOSL through reasoning rules and to generate DPV risk triples explicitly linked to their contractual origin, thereby targeting for an explainable risk assessment model. The two main contributions of this work are (i) a proposal for the modelling of risks associated with potentially abusive terms and (ii) the integration of ODRL/TOSL with the DPV to infer risks from rules.

## 2. Background and Related Work

Consumer protection[1] has long been a central priority of the European Union, as evidenced by the numerous directives, regulations, and initiatives adopted over the last few decades. It began with the Unfair Contract Terms Directive (UCTD), designed to protect consumers from unfair terms in unilaterally drafted contracts, and was subsequently complemented by additional consumer protection directives, such as the Unfair Commercial Practices Directive (UCPD), which addresses misleading and aggressive commercial behaviour, and the Consumer Rights Directive (CRD), which protects pre-contractual transparency and harmonised rights.

In October 2024, the European Commission published the final version of the fitness check of EU consumer law on digital fairness, assessing the effectiveness of these three directives in achieving EU policy objectives [5]. The evaluation identified persistent structural challenges, particularly in the digital environment, including power imbalances, the continued use of unfair contract terms, manipulative interface designs, and the growing complexity of digital products and business models.

Prior research has focused primarily on the automated detection of unfair contract terms in ToS and on explaining why such clauses are considered potentially unfair [1, 7, 8, 9, 10]. Complementarily, several studies have addressed the detection of dark patterns in online services and have categorized the risks posed by behavioural manipulation [11, 12, 13].

Formal vocabularies and policy languages have been developed for representing normative constraints in a machine-readable form. The DPV provides a foundational framework of common concepts for the interoperable representation and exchange of information about processing personal and non-personal data, as well as related technologies such as cloud services and AI [14]. DPV also includes extensions to its core vocabulary, such as the risk extension, which introduces terms for describing threats, vulnerabilities, impacts and consequences related to data processing. The ODRL, a W3C recommendation, is an ontology for expressing policies that specify permissions, prohibitions, and obligations [15, 16]. In this context, the TOSL profile is an ODRL extension that has been proposed to represent ToS clauses by incorporating specific elements of online contracts, such as jurisdiction, applicable law, and liability, thereby enabling automated analysis tasks such as the detection of potentially unfair terms [17, 18].

## 3. From Unfair Terms to Risks

Unfair terms are defined by the UCTD as contractual terms that (1) have not been individually negotiated and (2) are contrary to the requirements of good faith, causing a significant imbalance in the parties' rights and obligations to the detriment of the consumer [4]. While the UCTD establishes the legal criteria for unfairness, it does not articulate the risks that such imbalance may generate in practice.

To make this risk explicit, this section maps the identified types of unfair contract terms to the risks they generate for consumers. Building on the taxonomy introduced in Table 1, we analyse how each

---

[1]https://commission.europa.eu/law/law-topic/consumer-protection-law

potentially abusive term type has been associated with consumer detriment in Lagioia et al. [19] and in the European Commission's Fitness Check, Annex VI, Section VI.1.6 (Unfair contract terms) [5]. In the former, they analyse the reasons of the abusive terms found in ToS and identify specific forms of consumer detriment associated with five term categories. The EU report assesses whether the current framework still protects consumers and highlights risks for each type of clause. The objective of this mapping is to systematize the reported risks associated with each unfair term type, as set out in Table 1, and to provide a structured basis for their subsequent formalisation using DPV risk concepts, as presented in Table 2.

**Table 1**
Reported risks associated with unfair contract term types.

| Unfair Term Type | Risks (Lagioia et al. [19]) | Risks (Fitness Check [5]) |
|---|---|---|
| Arbitration | Awareness; cost; secrecy; limited remedies and judicial review | Excluding or hindering the consumer's right to take legal action |
| Choice of Law | – | Exclusion of legal rights |
| Content Removal | Financial loss; damage to private sphere; risk of being overpowered | Loss or damage of consumer data; prohibition to recover stored data; omission of information |
| Contract by Using | – | Financial harm; hidden or inaccessible terms |
| Jurisdiction | – | Excluding or hindering the consumer's right to take legal action |
| Limitation of Liability | Deprivation of remedies; difficulty recovering damages; data, reputation and economic loss | Placing the entire risk on the consumer; no compensation; loss of data or service |
| Unilateral Change | Uncertainty; lack of notice; serious harm from unanticipated changes | Legal uncertainty; lack of notice; exclusion of legal rights |
| Unilateral Termination | Consumer detriment; provider discretion | Consumer detriment; loss of access to the service or the account |

While limiting the legal rights of consumers by requiring arbitration or dispute resolution under foreign law or outside their country of residence is inherently risky, such clauses may also generate additional harms. These include loss of consumer confidence due to unfamiliarity with the mechanisms, language, or even the applicable legislation, potentially requiring legal representation in the foreign jurisdiction, as well as increased dispute resolution costs. Similarly, content removal terms not only authorize the provider to unilaterally delete a user's content or material without prior notice or reason, but also give rise to risks of personal or non-personal data loss or damage. As a result, consumers may experience a sense of powerlessness and could also incur financial losses.

Contract by using clauses infer consumer consent from the use of the service. As a result, such clauses generate risks related to the limitation or obstruction of consumer rights, and they could affect users' financial losses and non-material damage. In addition, limitation of liability clauses may dismiss the provider's accountability, thereby placing the entire risk on the consumer. This issue becomes even more problematic when the provider does not establish any form of compensation in cases of data or service loss, potentially resulting in economic harm.

Unilateral change and termination clauses create significant uncertainty for consumers, who may not know whether contractual terms, features, or prices will be altered, or whether the service may be discontinued altogether. Providers typically limit their responsibility to notifying users or requesting renewed acceptance of the modified terms, often without providing sufficient justification or safeguards. Such clauses may undermine the legal rights of consumers and may result in serious harm, including loss of access to accounts, functionalities, or continuity of service.

The DPV risk extension[2] provides a structured vocabulary of risk concepts that enables a formal

---

[2]https://w3c.github.io/dpv/2.2/risk/

**Table 2**
Mapping unfair term risks to candidate DPV risk concepts.

| Unfair Term Type | Candidate DPV Risk Concepts |
|---|---|
| Dispute Resolution (Arbitration, Choice of Law and Jurisdiction) | risk:InabilityToEstablishLegalClaims · risk:RightsDenied risk:RightsLimited · risk:RightsObstructed · risk:JudicialCosts risk:RightsExercisePrevented · risk:LegalSupportLimited |
| Content Removal | risk:DataLoss · risk:DataUnavailable · risk:DataErasureError risk:FinancialLoss · risk:NonMaterialDamage risk:CustomerConfidenceLoss |
| Contract by Using | risk:RightsLimited · risk:RightsObstructed · risk:FinancialLoss risk:CustomerConfidenceLoss · risk:NonMaterialDamage |
| Limitation of Liability | risk:FinancialLoss · risk:DataLoss · risk:ReputationalLoss risk:ServiceLimited · risk:DataUnavailable · risk:Harm risk:RightsDenied · risk:RightsLimited |
| Unilateral Change and Termination | risk:ServiceTermination · risk:DataUnavailable · risk:Harm risk:QualityDegraded · risk:CustomerConfidenceLoss risk:NonMaterialDamage · risk:FinancialLoss risk:RightsLimited · risk:RightsObstructed |

semantic representation of the identified unfair term consumer risks identified in Table 1. The mapping of these risks is aligned with the closest corresponding DPV concepts. When a direct correspondence exists, such as data loss or financial loss, the mapping is straightforward using `risk:DataLoss` and `risk:FinancialLoss`. For broader legal notions, such as exclusion of legal rights or consumer detriment, we operationalise them through combinations of DPV concepts, such as `risk:RightsDenied`, `risk:RightsLimited` and `risk:RightsObstructed`. The resulting mapping, presented in Table 2, provides a structured bridge between legal analysis and semantic modelling.

## 4. Representing and Detecting Risks in Policies with ODRL and DPV

The semantic representation of ToS can be formalised using TOSL, the ODRL profile specifically designed to model contractual clauses. This agreement representation supports the automatic detection of potentially unfair terms through SPARQL queries [17]. For instance, Listing 1 semantically represents an Amazon[3] content removal clause modelled with TOSL, in which the provider can delete the user account without prior notice and without providing a reason. This clause is identified as an unfair term because there is neither a duty to inform the customer in advance nor a constraint requiring justification for the deletion of the user account (as indicated in the comments in Listing 1). Thus, certain potential risks arise for the consumer, primarily the lack of availability or damage of data and the interruption of services at the provider's discretion.

Listing 1: TOSL representation of an Amazon content removal clause

```
1  # Potential mitigating constraints and duty
2  # - constraint: justification = valid_reason
3  # - constraint: delayPeriod >= X days
4  # - duty: provider informs customer
5
6  :PermissionDeleteUserAccount a odrl:Permission ;
7      dcterms:description "Amazon can delete your account without prior notice and without a reason" ;
8      odrl:assignee :provider ;
9      odrl:action tosl:remove ;
10     odrl:target :userAccount .
```

In line with the DPV risk structure, which follows the ISO 31000 standard, a potentially unfair rule constitutes a risk source that generates associated risks, defined as possible harmful events that may

---

[3]https://www.amazon.com/gp/help/customer/display.html?nodeId=GLSBYFE9MGKKQXXM

affect the consumer. The direct result of a risk materialising is the consequence, which generally affects the level of the system or service. When that consequence results in harm to the consumer, it is classified as an impact. Mitigation measures can be defined to address these risks. The definition of the risk related to the Amazon clause is presented in the Listing 2. The two main risks are data unavailability and service disruption, which may lead to immediate consequences such as data loss, and may ultimately result in significant impacts, including financial loss. Increased clarity and transparency in the contractual terms may serve as mitigation measures to reduce these risks.

Listing 2: DPV risk modelling of the potentially unfair Amazon content removal clause

```
1   ex:R1CR a dpv:Risk ;
2      skos:broader risk:DataUnavailable, risk:ServiceTermination ;
3      dpv:hasRiskSource [
4         a new:UnfairContractClause ;
5         dpv:hasContractualClause ex:PermissionDeleteUserAccount
6      ] ;
7      dpv:hasConsequence [
8         a risk:DataLoss, risk:ServiceNotProvided ;
9         dpv:hasImpact risk:FinancialLoss, risk:NonMaterialDamage, risk:CustomerConfidenceLoss ;
10        dpv:isMitigatedByMeasure new:AccountDeletionNotice, new:DeletionJustification
11     ] .
12
13  new:UnfairContractClause a dpv:RiskConcept, risk:PotentialRiskSource, risk:PotentialRisk ;
14     skos:broader risk:OrganisationalManagementRisk .
15
16  ex:PermissionDeleteUserAccount a dpv:Permission .
17
18  new:AccountDeletionNotice a dpv:OrganisationalMeasure, dpv:RiskMitigationMeasure ;
19     skos:broader dpv:Notice .
20
21  new:DeletionJustification a dpv:RiskMitigationMeasure .
```

Can the automatic detection of unfair terms be extended to the systematic inference of associated risks through rule-based reasoning? Once ToS are represented as structured, machine-readable policies using TOSL, identified risks can be systematically mapped and expressed using DPV, thereby improving transparency and facilitating risk assessment. Our vision is to develop a rule-based framework for the detection of potentially unfair terms in ToS and the inference of associated risks. The approach would rely on a set of formal rules designed to detect unfair term patterns, and to infer corresponding risk triples. These rules could be implemented and executed using the N3 reasoner [20]. The aim is to generate a ranked list of risks (from highest to lowest), along with an explainable justification (e.g., "This clause is risky because it lacks prior notice or justification requirements."). In addition, the system would enable consumer-centred risk assessment, allowing risk analysis based on consumer needs, such as requirements that data cannot be removed or that the agreement may change only with prior notice.

## 5. Conclusions

Unfair contractual clauses should be understood not only as legal irregularities, but as structural sources of contractual risk in contracts. This work explores how such risks can be formally modelled using the DPV and inferred through the integration of TOSL with a rule-based reasoning engine. The objective is to address a consumer-centred question: *what risks do we assume each time we accept Terms of Service?* To this end, the aim is to automatically generate a prioritised list of risks derived from potentially unfair terms. Future work will focus on implementing the N3 reasoning rules.

## Acknowledgments

This publication was supported by the R&D projects PID2021-126227NB-C21, PID2021-126227NB-C22, and PDC2022-133521-I00, funded by MICIU/AEI/10.13039/501100011033/ERDF/EU; and PID2024-155693NB-C44, funded by MICIU/AEI/10.13039/501100011033/FEDER, EU.

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
