# OpenReview forum: "Towards a Risk-Based Analysis of Terms of Service for Consumer Decision Support"
_SolidProject.org/SoSy/2026/Privacy_Session — SoSy2026-Privacy Paper_

### Official Review · ~Anelia_Kurteva1 · 2026-02-23
**Identifying risk in terms and conditions to empower users**

**Rating:** 6
**Confidence:** 5

**Review:**

The paper investigates the potential risks consumers face when signing up for different software services. The authors identify risks by analysing terms of service (ToS) for clauses that may be considered unfair. This was an interesting paper to read, and I look forward to its discussion at the conference.

Some concrete comments:
Regarding Table 1, could the authors clarify the target stakeholders for each identified risk? It would be helpful to include an additional column specifying the affected party, or to update the caption if they all pertain to the end-user. While risks like 'awareness' clearly impact the end-user, others seem more relevant to the organisation. Should we balance risks for end-users and for the organisations to ensure somewhat fair-play for both parties?

Additionally, I am wondering how this compares to using an LLM prompted with keywords/unfair terms to extract or detect issues within terms and conditions documents? If the goal is to adopt this approach at scale, assist end-users and empower them, how will this approach compare to simply using LLMs for document analysis? This is what most users are doing either way currently (e.g. asking an LLM to summarise complex documents and derive the most important points). Still, LLMs are not that good at this task and hallucinate, especially for legal texts. Do you envision a combination or extension of this approach with LLMs?

It is a bit unclear how this approach will help consumers since there is no discussion about how the approach will be available to them. Will it be a feature provided by SOLID, or a user interface?

Finally, the connection to Solid pod providers and the broader theme of decentralisation becomes less clear (on paper) as the paper progresses. Reinforcing this link in the later sections would strengthen the narrative. Does this approach apply to only decentralised terms and conditions, or could it be generalised to also centralised software providers?

Some inconsistent capitalisation in the references.

---

### Official Review · ~Cesar_Augusto_Fontanillo_López1 · 2026-02-27
**Accept, risk conceptualisation nuances**

**Rating:** 8
**Confidence:** 3

**Review:**

The submission is clear and is specific in the objectives it aims to achieve. It attempts to develop a rule-based framework for
The detection of potentially unfair terms in ToS and the inference of associated risks.

The introduction builds naturally and is well documented. Perhaps a possible question that the authors may ask themselves is why they chose that selected definition of risk in relation to their research objectives. Risk has been understood recently as 'potential negative consequences' to the rights and freedoms of individuals (Kloza, D’hulst and Aouadi). The concept of risk of the authors is a different one, as it also includes situations that would make it more difficult to 'achieve objectives'. Several questions arise:

1. With respect to whom are those obstacles to be evaluated, with respect to data subjects, organisational risk, societal, all?
2. What effect does this concept of risk have on the research objective beyond other risk conceptualisations?
3. What does it add or neglect?

I raise these questions for your own reflection, but also in light of the conflicting definitions of risk that underpin the submission, i.e. DPV,  used as part of the research, seems to follow a different definition (as you write: 'possible harmful events that may affect the consumer'), which may be more limiting than your own. Are there specific effects in terms of identified risks that go beyond DPV, which you consider risks under your own conceptualisation? If so, which and why?

The background and problem statement seem to be also well developed. The paper seems to go a step further than other projects and infers the concrete risk on the basis of rule-based reasoning, with different levels of risk associated. Several questions may be asked to this extent:

1. How would the risk ranking be made? I understand you will use risk extension as a source. Do you plan to use specific scales? Which ones? If so, why so?
2. PDV terms will be used. However, it is not quite evident to me in PDV what the differences are between risk RightsDenied, RightsEroded, RightsExercisePrevented, RightsImpact, RightsLimited, RightsObstructed, RightsUnfulfilled, RightsViolated. It seems that many of these categories overlap. I understand that this does not directly concern your own initiative, as you just use the existing vocabulary. However, perhaps worth explaining why each category is used and why it would be considered an autonomous risk category that adds something. Same for services and other potential categories you may use. I think the research would benefit from justification at that level, especially for legal scholars.

---

### Decision · Program_Chairs · 2026-03-09

Accept (Paper)